# SimpliPyTEM: An open-source Python library and app to simplify transmission electron microscopy and *in situ*-TEM image analysis

**Gabriel Ing**[1]*, **Andrew Stewart**[2], **Guiseppe Battaglia**[3,4], **Lorena Ruiz-Perez**[3]*

**1** Institute of Structural and Molecular Biology, Department of Chemistry, University College London, London, United Kingdom, **2** Department of Chemistry, University College London, London, United Kingdom, **3** Institute for Bioengineering of Catalonia, The Barcelona Institute of Science and Technology, Barcelona, Spain, **4** Catalan Institution of Research and Advanced Studies, Barcelona, Spain

* gabriel.ing.19@ucl.ac.uk (GI); lruiz@ibecbarcelona.eu (LR-P)

## Abstract

Introducing SimpliPyTEM, a Python library and accompanying GUI that simplifies the post-acquisition evaluation of transmission electron microscopy (TEM) images, helping streamline the workflow. After an imaging session, a folder of image and/or video files, typically containing low contrast and large file size 32-bit images, can be quickly processed via SimpliPyTEM into high-quality, high-contrast.jpg images with suitably sized scale bars. The app can also generate HTML or PDF files containing the processed images for easy viewing and sharing. Additionally, SimpliPyTEM specifically focuses on *in situ* TEM videos, an emerging field of EM involving the study of dynamic processes whilst imaging. The package allows fast data processing into preview movies, averages, image series, or motion-corrected averages. The accompanying Python library offers many standard image processing methods, all simplified to a single command, plus a module to analyse particle morphology and population. This latter application is particularly useful for life sciences investigations. User-friendly tutorials and clear documentation are included to help guide users through the processing and analysis. We invite the EM community to contribute to and further develop this open-source package.

**Data Availability Statement:** Software developed can be found in a public repository on github, accessible at https://github.com/gabriel-ing/SimpliPyTEM, documentation and tutorials can be

## Introduction

Electron microscopy (EM) is a powerful technique for observing samples at the nanoscale, and it is unrivalled for ease and popularity of use [1]. As with most modern-day microscopy methods, EM imaging nowadays yields data in the form of digital images or videos, or arrays of numbers, with high and low values representing bright and dark regions of the image respectively. In conventional, bright-field EM, the intensity of the signal corresponds to the density of the sample at each point, with more dense regions transmitting fewer electrons and thus yielding dark regions in the images. There is considerably more data available in the average EM image than can be seen with the naked eye, for example, the images are often 16-bit or 32-bit allowing for far more contrast than is displayed. Combined with this, there is often a

found on readthedocs at https://simplipytem. readthedocs.io/en/latest/. The software can be installed through PyPI from here: https://pypi.org/ project/SimpliPyTEM/.

**Funding:** GI acknowledges the Wellcome Trust for funding his studentship (222908/Z/21/Z). The funders had no role in study design, data collection and analysis, decision to publish, or preparation of the manuscript.

**Competing interests:** The authors have declared that no competing interests exist.

considerable amount of noise in EM images, arising from various sources including from inelastically scattered electrons and electrons scattered due to multiple collision events [2], alongside undesired detector and readout signals. Noise can substantially obscure the underneath image. However, noise can often be reduced or removed using techniques ranging from simple linear filters [3] to complex methods, including deep learning-based methods [4–7]. Effective data processing of EM images is thus vital to maximising the information yielded by electron microscopy experiments. This requirement is even more significant in the world of *in situ* EM, where the ability to observe the behaviour of materials and nanoparticles under real-time, and controlled conditions can provide novel insights into advanced materials and novel nano-structures. *In situ* TEM typically generates a significant amount of data in the form of videos thus effective data processing and post-acquisition workflow optimisation become essential tasks.

With these requirements considered, a lot of specialised and manual image processing work is often required for post-experimental analysis. However, much of the post-experimental process cannot be easily automated, as the files produced are commonly incompatible with standard image viewing programs, they often have poor contrast and lack scale bars. As a result, many users spend significant time performing basic post-imaging tasks including contrast enhancement, basic filtering and adding scale bars. These tasks can be automated. It can also be time-consuming to examine the acquired images as a whole or simply to produce a presentation to view and share these, for example for discussing the results with colleagues. Therefore, EM users are in constant need of methods to automate the basic data processing steps and allow the production of experimental contact sheets. Combined with this, the possibility to access the metadata from the images collected is also important, so users can quickly access details about images. However, metadata is often hidden within files and not easily accessible, making image curation a time-consuming task.

Many programs are available to process EM images, with one of the most common approaches being the use of ImageJ [8], an open-source package for scientific image analysis. ImageJ effective for manually editing images and has a scripting-based macro language to automate repetitive tasks. However, the scope for automating complex tasks is limited and many of the most powerful image analysis tools are not available with ImageJ. Gatan's Digital Micrograph program [9] is also effective at various image analysis tasks, however is only available on Windows operating systems. Coded approaches can be very effective, and the most popular language since October 2022, is Python [10], Python is also widely used for science and image analysis. Many image analysis libraries and functions are available with Python, including openCV [11], pillow [12] and scikit-image [13], making it possible to perform an enormous variety of tasks. One advantage of using Python for this analysis is the prevalence of Python-based machine learning and deep learning-based image analysis tools [14, 15], which commonly use images in a similar format and can easily be combined with other Python-based workflows. At the same time, the range of available options and locations can make it difficult for newcomers to locate the required methods. Many of the functions in these libraries also come with many parameters which can make the function much more complicated to use than is necessary for most cases. Whilst Python can be difficult to learn, the wide range of users and tutorials available freely online can aid the learning process. While scripting tends to be more powerful than user-interface-based approaches due to the wider availability of options, many potential users find coding intimidating and challenging to learn, leading to users performing time-consuming manual methods.

Herein, we introduce a new app for basic image and video processing, and visualisation. In addition, we also introduce a Python library for work of added complexity. The app aims to allow effective image processing from large file-size images or video files in various formats to

reduced-size, high-contrast JPEG files with scale bars. HTML or PDF documents containing these images and videos can also be created for easy visualisation. The Python library aims to build on available methods by lowering the barriers of users new to Python for analysing EM images and videos. An extensive range of functions, including image and video visualisation, filtering, contrast enhancement and scale-bar addition are available in a user-friendly, consistent and well-documented manner, with few required parameters. This package is accompanied by detailed documentation and iPython notebook-based tutorials to make it easy to access and follow for users with only basic knowledge of Python.

## Results

### SimpliPyTEM-library for images

Python is among the most powerful and widely used image processing and image analysis tools. There are lots of modules available for image analysis, which can be very effective, however many of these modules involve steep learning curves and unnecessarily large numbers of parameters. Therefore, to make common methods more available, we have created a library drawing upon some commonly used methods, including openCV, scikit-image and numpy [11, 13, 16]. This library is built on the principle of making the methods simple to use while sacrificing little performance. **Fig 1A** and **Table 1** display some of the available functions within the library. In contrast, **Fig 1B** shows a code snippet, demonstrating the simplicity of using the code, and **Fig 1C** shows the effect of the code snippet on a single example image. This code ran in less than 2 seconds on a MacBook Pro, 2018 for a 32-bit 3838x3710 pixel image, demonstrating the speed at which the processing can be performed.

SimpliPyTEM image processing is primarily hosted in a single Python class called 'Micrograph'. The Micrograph object hosts the image data, metadata and pixel size, and the methods to process the image. The library currently contains many simple methods to process the images, including image filtering (with median, gaussian, low-pass, non-local means and Wiener filters), the addition of scale-bar, converting to 8-bit, contrast enhancement and extracting metadata from digital micrograph images. These functions are all designed to be performed with a single line of code and with as few required parameters as possible, making the functions as easy as possible. These functions return a copy of the object, meaning the original object is kept.

In addition to conventional image processing methods, SimpliPyTEM includes support for denoising images and videos using the deep-learning based denoising method Topaz, produced by Bepler *et al.* [6]. This method uses a deep learning model trained to predict and remove noise from cryo-EM images by comparing two different examples of noisy cryo-EM images. Topaz can dramatically enhance images and videos, particularly by making low resolution features much easier to see. Although Topaz is specifically trained for cryo-EM, it can be highly effective for various bright-field TEM images and videos including *in situ* TEM videos. Unfortunately, this does have significant hardware requirements and benefits greatly from a CUDA GPU for fast processing. On a Macbook Pro, 2018 Topaz takes several minutes per 32-bit 3838x3710 pixel image, while a CUDA GPU can accelerate this to mere seconds. By integrating Topaz within SimpliPyTEM-GUI, we aim to make state-of-the-art denoising methods more accessible and user-friendly for any researcher.

### SimpliPyTEM-library for videos

*In situ* TEM is a growing field, allowing the capture of live nanoscale events, providing dynamic information on phenomena that are not easily studied using other methods [17]. This results in EM videos containing a lot of information, particularly if captured from a direct

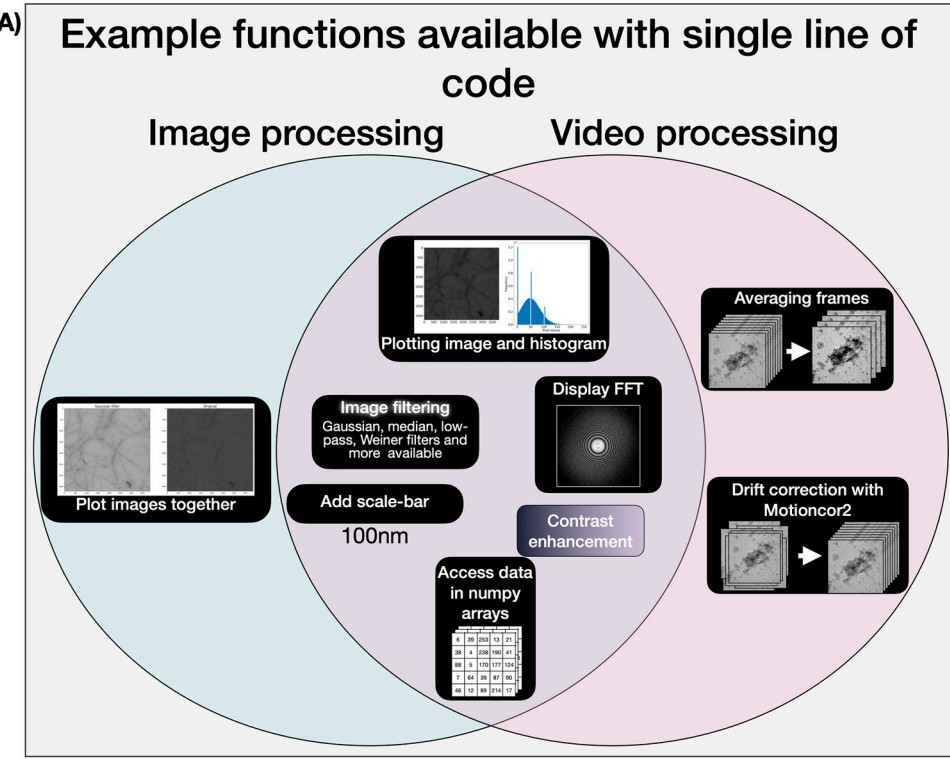

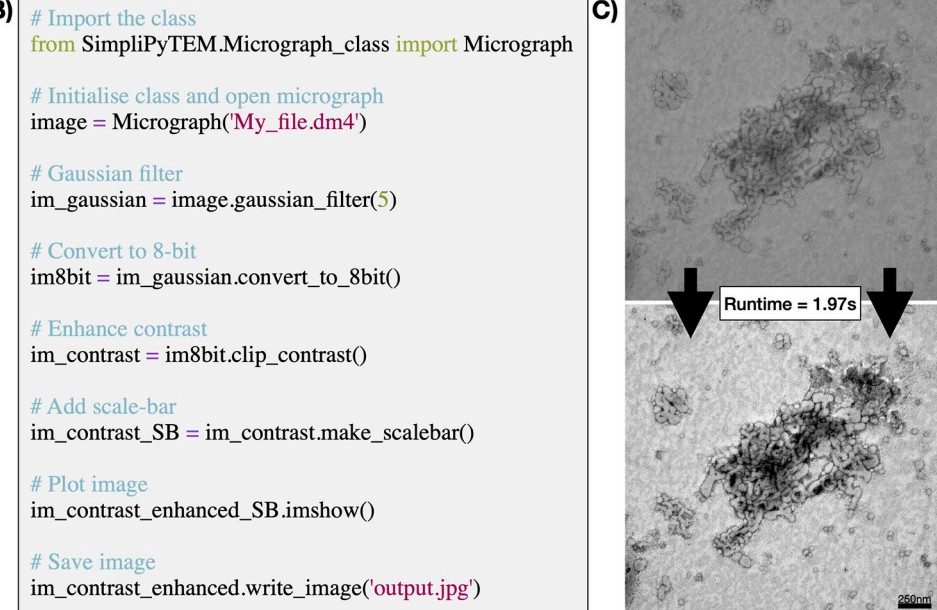

**Fig 1. SimpliPyTEM—Python library. A)** Venn diagram showing an example of available functions for image and video processing, these are all accessible with a single line of code. **B)** Example code for basic image processing. **C)** Image transformation achieved by the code shown in B, with a running time of 1.97s, with it falling to 0.84s when the image is not plotted (on MacBook Pro, 2018).

electron detector, which commonly have high bit-rates and large sizes. Efficient analysis of acquired videos is essential, and as discussed in the introduction Python is ideal for this, however can prove difficult for inexperienced users. As such, we have created a Python library

**Table 1. List of functions available for image and video processing.**

| Function | Image | Video |
|---|---|---|
| Contrast enhancement | | |
| Median Fillter, gaussian Filter | | |
| Wiener filter, Low-pass filter, non-local means filter | | |
| Display Fourier Transform | | |
| Bin in x and y dimensions | | |
| Crop | | |
| Improve uneven contrast | | |
| Show in Jupyter notebook | | |
| Save key metadata into.csv file (from DM files) | | |
| Save into PDF | | Can save average frame |
| Save into HTML | | |
| Save as 8-bit TIF, 32-bit TIF, JPEG | | Can save average frame. Can save choice of frame though coded approaches. |
| Save videos as TIF stack, TIF sequence, MP4 or AVI video files | N/A | |
| Average frames | N/A | |
| Correct for motion using motioncor2 | N/A | |
| Denoise using Topaz Deep-learning denoiser | | |

The table shows the functions available with both the code library and the GUI (Green) whereas the purple shows functions only available using the code library.

allowing for many basic and advanced methods to be employed for analysis of videos. Such methods include all of the techniques discussed for the image library, while also including many video-specific methods, for example, averaging frames together, video normalisation, and using existing software, motioncor2 [18] to correct for global motion within the video.

Videos can be loaded into a 'MicroVideo' object from various sources, including DM files, movie files like MP4, AVI, MOV and sequences of TIF or DM image files. From here, the videos can be averaged into groups of n frames, averaged in a sliding window fashion, converted to 8-bit, contrast enhanced and filtered, added a scale-bar, alongside several other functions. As with the image-processing library, these functions can be achieved in single lines of code, with few required parameters to make them as simple as possible. By processing videos in this way, the user can easily prepare the video for further analysis or presentations. Moreover, the video can be easily viewed in an iPython notebook (e.g., Jupyter notebook) and saved as an image sequence, an image stack, a single image (either single frame or average) or a movie file in.mp4 or.avi format, depending on its intended use.

## SimpliPyTEM-GUI

The GUI-based image processing app is designed as a simple tool for EM users to use during or post-experiment (Fig 2). Images and videos in a number of common formats, including digital micrograph (GATAN),.emi (FEI) and.TIF can be loaded, enhanced, and saved into.JPG or.TIF images. Basic processing tasks can be performed on the images, including gaussian and median filters, the addition of a scale bar and contrast enhancement. Different options are also available for videos, including DM image stacks,.mp4 and.avi files to be saved as an average image, a video (.mp4 or.avi), a motion-corrected average (using motioncor2), a tif sequence (i.e. saved as individual tif files) or a tif stack. This app provides many options for *in situ* users

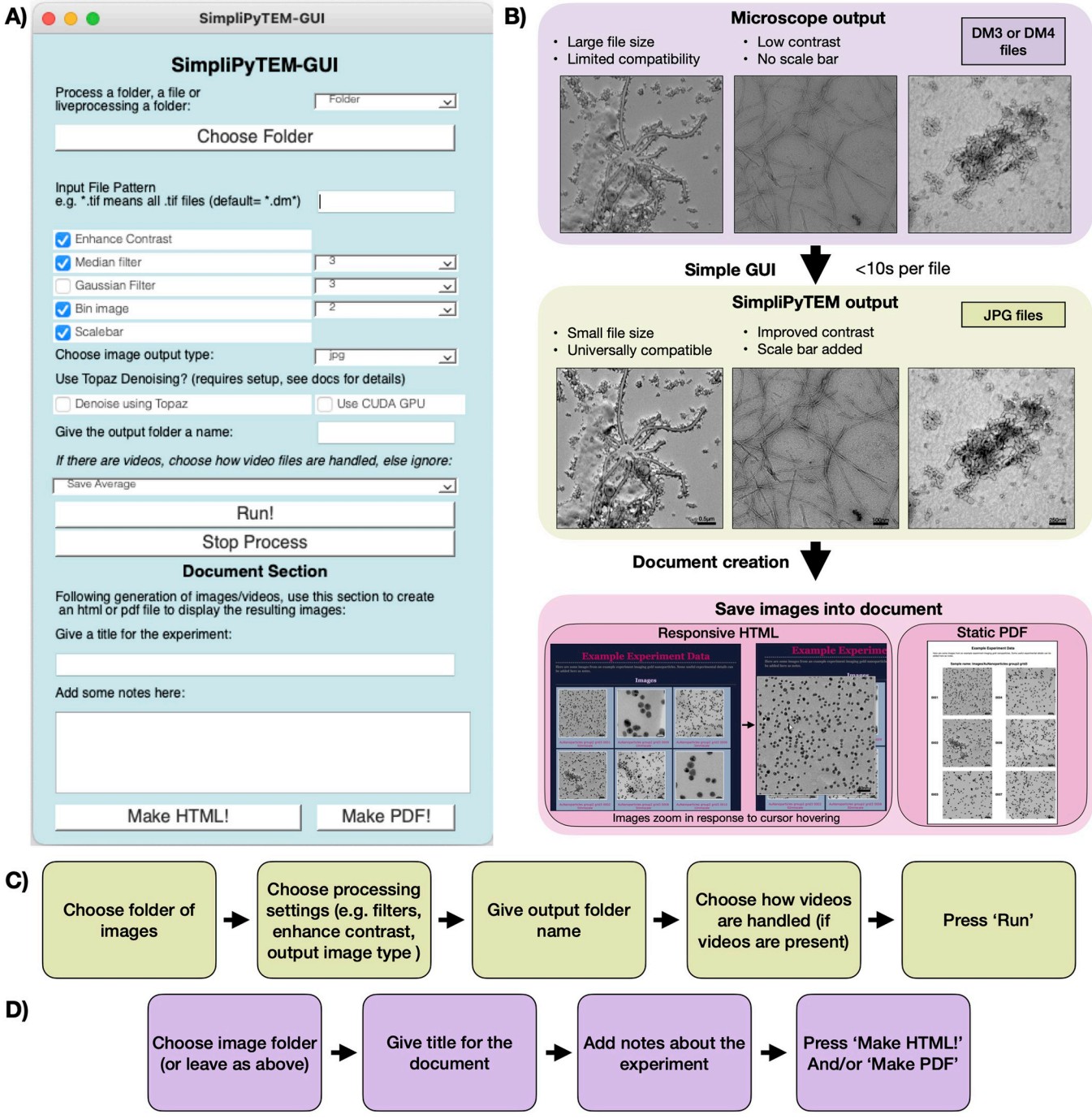

**Fig 2. SimpliPyTEM-GUI for post-acquisition image processing.** A) Appearance of SimpliPyTEM-GUI. B) Effect of SimpliPyTEM-GUI, showing the simple conversion of digital micrograph files with poor contrast and limited compatibility, to high contrast JPGs which can be used for observation and display. This process is quicker than many comparable methods, including using imageJ, taking seconds per file. There is also a document creation section which allows a PDF or responsive HTML document to be produced showing the images and videos collected during the experiment. Examples of these documents are shown at the bottom of B). C) A flowchart showing the steps takn to use the GUI to process a folder of images. D) A flowchart showing the steps taken to produce an HTML or PDF document containg the required images.

to create effective previews of their videos. The functions accessible from the GUI are detailed in Table 1. Flowcharts for how the GUI can be used are included in Fig 2C and 2D.

Alongside the image or video processing, there is also an additional section for visualisation. Images can be easily added to PDF or HTML files with a designated title and experimental notes. This process allows the rapid generation of documents to summarise the results of an experiment, which can be conveniently viewed or shared with others with very little preparation time. This development is beneficial as it allows for rapid sharing and discussion of the results with colleagues or collaborators. Examples of these documents can be seen in Fig 2.

As discussed in the introduction, image metadata can be useful for various reasons, for example to easily get an idea of the magnification used in an image. This value may provide information about what the acquired image contains without the need to view the image itself. SimpliPyTEM-GUI will automatically extract many of the features found within these metadata tags, including magnification, voltage, exposure time and acquisition date and time. These values are collected into a CSV table with the other files within the folder, which can then be easily examined, allowing easy identification of files by their imaging conditions.

## Particle analysis module

While the basic image and video processing modules, Micrograph_class and MicroVideo_class, are highly effective for image processing, we also include a basic image analysis module. The module contains simple methods for extracting data from nanoparticle, including positions, sizes, morphology, shape, and other physical properties. This information is crucial for thorough sample characterisation and optimisation in various fields, including pharmaceutical and materials science, nanotechnology, and biomedicine. The process for measuring particle properties is simple and involves applying a threshold to separate the nanoparticles from the background intensity, locating the boundaries of the particles and filtering the selected particles by area. The process will yield the user to measure the physical characteristics of the particles, which are returned in a Python dictionary of characteristics. These physical properties include area, position, circularity, major and minor axes, and major: minor axis ratio. The module also includes a way to take multiple measurements of the particle diameter from a single particle, allowing the diameter's maximum, minimum, mean, and standard deviation to be collected. By considering these measurements, the user can obtain quantitative information about the uniformity of the investigated particles. An example of this functionality is given in **Fig 3**, where a negatively stained micrograph of polymer particles was selected for applying this module. Thresholding, was applied to the image, then the objects in the field of view, i.e. particles were located, filtered by size, and various properties could subsequently be extracted and plotted.

## Detailed methods

### Image manipulation

The main aim of this package is to simplify the use of Python-based programming for processing and analysing electron microscopy images. Python already has many publicly available libraries for image analysis [11–13], some of which have been used to varying degrees in this package. Image processing with Python is commonly performed by holding images in a numpy array [16], these are n-dimensional matrices of values which store data and allow many efficient data-editing functions. These numpy arrays are the backbone of image handling and manipulation in SimpliPyTEM.

SimpliPyTEM supports the opening of various image and video format files. Digital micrograph (DM) [9] files, a common EM file format from Ametek (formally Gatan) detectors, are

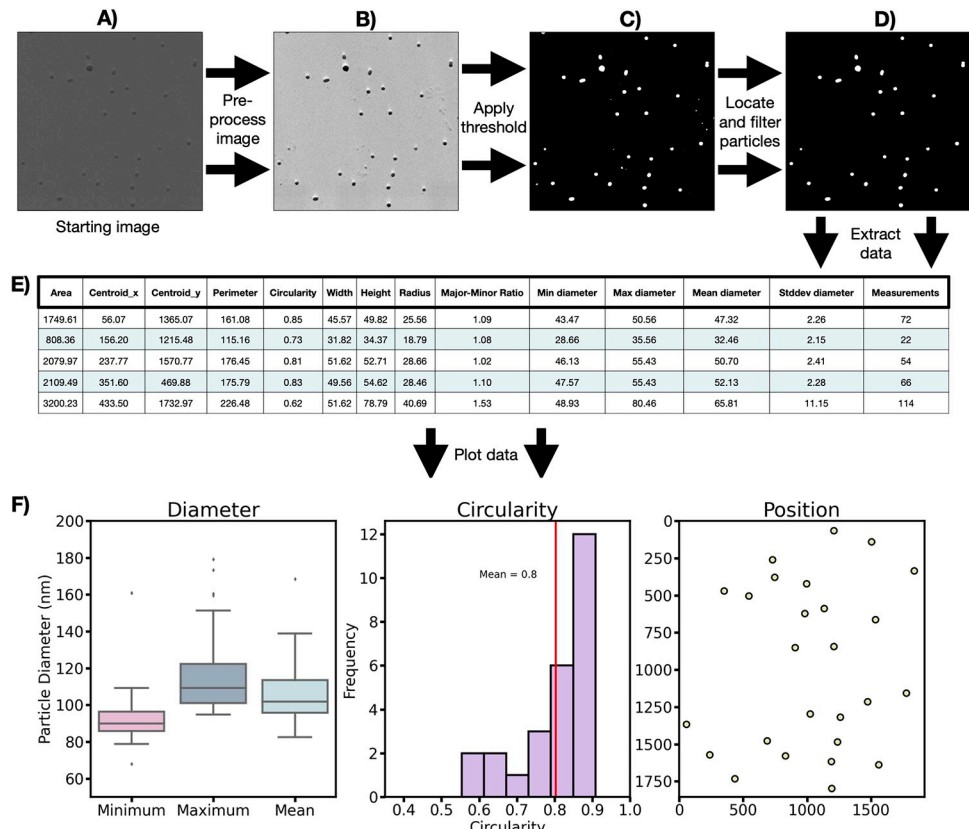

**Fig 3. Basic particle analysis protocol to extract particle data from a negatively stained image of polymer particles.** The starting image is opened (A), preprocessed to enhance contrast and features (B), then a threshold is applied to create a binary image, with particles and background in white and black, respectively (C). The particles are located and filtered by size (D), particles touching the edges of the image are also removed. Finally, data is extracted (E) and plotted (F) with very few lines of code (from image to data). Here we show plots of particles' diameter, circularity, and position, however more features are also accessible. The code to produce this analysis is available online as a tutorial within the documentation (https://simplipytem.readthedocs.io/en/latest/Tutorials/Particle_analysis_tutorial. html).

opened using openNCEM [19]. Metadata from DM files are extracted and easily accessible. Another popular EM image format, MRC files, are opened using the Python package mrcfile [20]. Both of these file types can include both images and movie files. Other Microscopy image files can be opened using the multi-format spectral analysis program Hyperspy [21], which allows a wide range of Micrograph formats to be opened, including those used by FEI, JEOL and Gatan.

In-situ TEM experiments are often recorded with screen recording software due to insufficiencies in direct and charge-coupled detectors for capturing videos. However, the data produced often require similar processing to detector-captured videos. Here, we support the opening of most major video formats, including MP4, MOV and AVI, these file types are opened using openCV's [11] VideoCapture module.

Several choices are available for outputting image and video files from the software. For images, there is a choice between TIF files and JPEG files. TIFs allow image data to be saved in the current conditions, thus producing uncompressed images or images with higher bitrates. The Python package tifffile is used for this task. JPEG files are also supported, in this case files are compressed, producing a much smaller file size. Saving images as JPEGs can introduce

subtle distortions in the image, but can be useful for ensuring files remain lightweight for easy viewing and sharing. The generated images are high quality nonetheless and ideal for display purposes.

Movie files have many export options, allowing for various downstream applications. Movie frames can be saved as sequences of TIF Files or as a TIF format image stack. Image sequencing can be particularly useful when investigating a dynamic process via *in situ* EM. MP4 and AVI movie files are saved using moviePy; MP4 files are effective for viewing and displaying movies, in particular, the MP4 format was chosen for its suitability to be displayed on webpages. The AVI files produced are uncompressed or raw, this format was chosen to be compatible with ImageJ.

### Image filtering

Several common image filters are used in this library. OpenCV [11] was used to implement median and gaussian filters. These filters use 2D convolution to reduce the noise levels within the image and are commonly used when viewing noisy images. Additionally, a non-local means filter is also introduced from OpenCV. Other 2D image filters included that aim to reduce noise within the images, such as a Wiener filter, here implemented using SciPy [22]. Low-pass filtering has been implemented using a number of numpy functions to perform fast Fourier transforms, along with OpenCV to produce a circular mask. Topaz denoising as implemented by Bepler *et al.* [6] can also be called via SimpliPyTEM. While these functions rely upon other libraries, they have been made more convenient to use by reducing unnecessary input requirements and simplifying the functions to call them. This simplification is particularly beneficial for video processing.

### Document generation

The project herein presented aims to create an automated method to view, share and present data collected by electron microscopy. To achieve this aim, we create PDF files containing all the images collected in an imaging session, this is a common format and easy to share as a standalone document. The Python package FPDF2 [23] was used to generate the PDF documents. Furthermore, to allow the user a more interactive experience, an HTML file containing all the images and movie files from an experiment can be generated. An accompanying CSS stylesheet is also produced to add to the interactive viewing experience. The HTML is generated with the Python library airium [24]. These options generate documents that can act like photography contact sheets, allowing users to rapidly view images and identify the images suitable for further processing or presentation.

### Displaying image and videos

The recommended usage method for SimpliPyTEM's Python library is to use an iPython notebook, for example, a Jupyter notebook [25]. These notebooks allow an interactive coding interface where images and plots can be displayed, and the underlying code can be easily edited and rerun. This method also allows new code to be run independently of the code which came before it, with variables still saved. To display images and plots within iPython notebooks, Matplotlib is used, while MoviePy is used to display videos.

### Contrast enhancement methods

The *clip_contrast* method is used to improve the contrast of an image or video by scaling the image to new white and black values. The advantage of this method, rather than other

examples like openCV's enhanced contrast or histogram equalisation, is that *clip_contrast* can provide reliable improvements without any user decisions, and thus can be used in automated pipelines. Image contrast is selected by entering maximum and minimum pixel values or a saturation value, the saturation here is the percentage of pixels above a maximum or below a minimum value. Hence the minimum and maximum values are selected using the saturation. The pixel values in the image are then scaled to the new minimum and maximum values, such that these values are between 0 and 255. This function therefore returns a contrast-enhanced version of the image or video, while the degree of this enhancement is controlled by the saturation value.

We have also implemented l*ocal_normalisation*, this aims to even the contrast out across an image, as often TEM images are bright in the centre and dark in the corners. This effect can be visually displeasing, but more significantly it can make image segmentation using thresholding challenging. This algorithm separates the image into n x n patches, and then scales these patches to the global median, such that each pixel in the patch is multiplied by the global-median / local-median. To reduce edge artefacts from the patches, padding can be used. Padding in this context is an overlap between adjacent patches, and the mean of shared pixels in overlapping patches is used in the final image. For this function, only the number of patches is required alongside the original image, and an image with more even contrast is outputted. An example of this method being used can be found within the documentation (https://simplipytem.readthedocs.io/en/latest/Tutorials/MicrographAnalysisTutorial.html#Fixing-uneven-contrast).

## Addition of scale-bar

A scale-bar can be easily added to an image with a single command. The size of the scale-bar is chosen as a fraction of the size of the image, meaning scale-bar sizes appear consistent for images of different dimensions. The pixel-size is taken from the metadata of DM files automatically but can also be loaded in when opening other files, or defined using a separate function (*set_scale)*. The colour is chosen to be either black or white based on whether the scale bar area has a significantly lower mean pixel value than the rest of the image. The pixel values in the specified position are changed using numpy. The scale-bar text is added using pillow [12], as this allows special characters, including 'μ', which is commonly used in scale-bars (for micron units: μm). The user can convert the scale between nanometers and microns with a single command, while other conversions can also be performed but do require a scaling factor or measurements to be included.

## Particle analysis

A basic particle analysis module is included within the package, this is designed to collect statistics on the particle morphology, including area, circularity, and maximum, mean and minimum diameters. The module includes methods to threshold particles, using tools available with openCV, and extract data from the particles into dictionaries or pandas databases. Such databases can then be used to plot figures in Python or export the data to CSV files.

Particles can be located by finding edges in the binary thresholded image, internal parts of the particle are filled in, and particles larger or smaller than set values are filtered out, particles on the edge of the image are also filtered out. Again these functions mainly use openCV and the Python package imutils. A labelled image can also be inputted to find morphology data, allowing users to locate particles using other available methods, for example object detection programs like StarDist [15]. Morphology data from particles are collected and returned as a Python dictionary. An additional option to measure each particle across many positions is also

included. In order to perform this measurement, every pair of coordinates is considered, and when these coordinates make an angle of $180°\pm1°$ with the centre point, the measurement of diameter is counted. This procedure was used in an image of polymer nanoparticles and shown in Fig 3 and can be found in more detail in a tutorial within the library's documentation (https://simplipytem.readthedocs.io/en/latest/Tutorials/Particle_analysis_tutorial.html).

## Metadata

Image metadata can be useful but difficult to access with TEM files. As such, we have tried to make it more accessible by saving the image or video metadata from DM files into a comma-separated value (CSV) file, showing many of the key values saved within the file. The metadata is taken from the headers of DM files using openNCEM [19], from which several key values are extracted. By doing so, a user could easily check which images were collected at a certain magnification, or when the images were acquired by looking at an automatically generated table. Unfortunately, at present, this task only works with DM files.

## Sample images

The sample images included in the manuscript were collected on a JEOL 2200 microscope with a Gatan K2 camera. These images include a range of stained polymer nanoparticles, amyloid fibres and gold nanoparticles.

## Conclusion

Herein, we present a new computational Python package to aid with the processing and analysis of image and video data from electron microscopy experiments. The package is fully documented and supported by tutorials, aiming to make one of the most powerful image analysis tools more accessible to beginner users. The proposed package is particularly beneficial to *in situ* EM investigations where early data evaluation and post-processing can help users identify trends and correlations that may not be apparent from the raw data. In this fashion SimplyPy-TEM allows users to make informed decisions about experiment design, sample preparation, etc by providing a fast and thorough evaluation of the data collected in the imaging session. Post-experimental image processing times can be reduced to mere seconds per file, and user-friendly documents to present, evaluate and share the data can be generated rapidly. By examining simple preview images within these documents, one can rapidly find the images or videos of particular interest for further analysis or display. Ultimately the aim for SimplyPyTEM is to share commonly used methods and unlock the potential of our data analysis. This, in turn, will help accelerate the science of all electron microscopy and *in situ* electron microscopy.

## Code availability

All the code is fully open-source and licensed under the GPL-3 licence. It can be downloaded and installed using Python's package manager 'pip' ('pip install SimpliPyTEM'), with the pyPI page for the package being found at https://pypi.org/project/SimpliPyTEM/. The code is also deposited on GitHub at https://github.com/gabriel-ing/SimpliPyTEM, documentation and tutorials can be found at https://simplipytem.readthedocs.io/en/latest/index.html.

## Acknowledgments

We would like to thank Valentino Barbieri, Chiara Cursi and Barbara Yus-Ibarzo, for providing samples which have become example images in this manuscript.

## Author Contributions

**Conceptualization:** Gabriel Ing.

**Software:** Gabriel Ing.

**Supervision:** Andrew Stewart, Guiseppe Battaglia, Lorena Ruiz-Perez.

**Writing – original draft:** Gabriel Ing.

**Writing – review & editing:** Andrew Stewart, Lorena Ruiz-Perez.

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
