## [Decision Letter · Decision Letter 0]

23 May 2023

PONE-D-23-11327SimpliPyTEM: An open-source Python library and app to simplify Transmission Electron Microscopy and in situ-TEM image analysis.PLOS ONE

Dear Dr. Ing,

Thank you for submitting your manuscript to PLOS ONE. After careful consideration, we feel that it has merit but does not fully meet PLOS ONE’s publication criteria as it currently stands. Therefore, we invite you to submit a revised version of the manuscript that addresses the points raised during the review process.

We look forward to receiving your revised manuscript.

Kind regards,

Carlos Fernandez-Lozano, Ph.D

Academic Editor

PLOS ONE

Journal Requirements:

   "GI acknowledges the Wellcome Trust for funding his studentship (222908/Z/21/Z). The funders played no further part in the research."

   "We would like to thank Valentino Barbieri, Chiara Cursi and Barbara Yus-Ibarzo, for providing samples which have become example images in this manuscript. GI acknowledges the Wellcome Trust for funding his studentship (222908/Z/21/Z)"

   "GI acknowledges the Wellcome Trust for funding his studentship (222908/Z/21/Z). The funders played no further part in the research."

Additional Editor Comments:

On behalf of the editorial team at Plos One, we would like to inform you that your manuscript has undergone a rigorous peer review process. We would like to express our appreciation for your contribution to our journal. The peer review process plays a pivotal role in maintaining the quality and integrity of scholarly publications. We are pleased to inform you that your manuscript has received thoughtful evaluation by our esteemed reviewers, who have provided their expert opinions, constructive feedback, and valuable suggestions for further improvement.

We kindly request that you carefully review the comments provided by the reviewers and consider them thoughtfully during the revision process. Each comment has been made with the intention of improving your manuscript, and addressing them will substantially enhance the scientific rigor and readability of your work.

Reviewers' comments:

Reviewer's Responses to Questions

**Comments to the Author**

1. Is the manuscript technically sound, and do the data support the conclusions?

Reviewer #1: Partly

Reviewer #2: No

2. Has the statistical analysis been performed appropriately and rigorously? 

Reviewer #1: N/A

Reviewer #2: N/A

3. Have the authors made all data underlying the findings in their manuscript fully available?

Reviewer #1: No

Reviewer #2: Yes

4. Is the manuscript presented in an intelligible fashion and written in standard English?

Reviewer #1: No

Reviewer #2: Yes

5. Review Comments to the Author

Reviewer #1: The manuscript presents a Python library dedicated to evaluate transmission electron microscopy (TEM) images. The Python library creates metadata files, enables image processing to reduce noise, add scale bar as well as enables analysis of particles. The innovation of this work is a complete toolbox via GUI or regular coding. The current manuscript should rewritten, addressing my concerns (see attached PDF document) before its publication in PLOS ONE.

Reviewer #2: This manuscript describes a simple package written in Python that is designed to make access to image stacks in various formats associated with current electron microscopic images. The purpose of the program is relatively simple - it is not for complex analysis, although it does some analysis, but for manipulating the images for sharing, perhaps preliminary analysis, and examination. I can see where this would be a useful tool for many labs.

One concern is that this paper does not describe scientific results, which is the first item in the list of elements for review, so it is not clear that it is a suitable fit for this journal. This is for the editors to determine. I think the work would be a stronger scientific contribution if the authors could show how it leads to or enables some scientific discovery.

The paper is clearly written, although in places it is somewhat repetitive - in particular the results text recapitulates some of the methods section. The documentation (on readthedocs.io) is fairly good, although there seems to be some disorganization (opening the "micrograph class" has a link to the "micro video class", which is actually the next element. the list of functions in the "Documentation" is a repeat of the "List of functions..." in the link above it. This should be cleaned up. The tutorials look nice however.

In general I am not a fan of using JPEG for image rendition, as it can introduce distortions, notably in the color space. It is probably ok for this particular use case, but the generation of output files should be flexible and include formats that do not involve the kinds of compression that can distort subtle aspects of an image. I think this should be empahsized more in the methods section.

line 82: Many languages are faster than Python (C and it's derivatives, and Julia, for example). Whether the code is accessed by scripting (command line) or from a gui is not really relevant here; the great advantage of a gui is user convenience and generality, and it should have little effect on the actual processing speed.

Code:

1. In several locations, "from xyz import *" is used. This is generally bad practice; use explict imports ("from xyz import abc", or "from xyz import abc as ABC"). Explicit is always better than implicit (Zen of Python).

2. The code presentation would benefit from some reformatting (using "black" and "isort").

3. The program uses PyQt5 as the gui backend. This has been superseeded by PyQt6. PyQt5 (as provided from the vendor) does not always work on Apple M1/M2 processors, and I was unable to get the code to even install using the instructions on readthedocs (macos, M1 processor). The documentation warns about this. It should not be hard to update this to PyQt6 to enable this functionality, given that most of the gui code appears to be in one fairly short file. This could extend the lifetime of the code, as well as its reach to other labs.

4. The requirements.yaml (and setup.py) files did not specify versions for most of the imported libraries. It can be useful to "pin" these to specific versions to ensure that the program will run in the future when the libraries might be updated with incompatible requirements or fixes to their code. For example, the documentation mentions Python 3.10, but the condo install instructions point to Python 3.8.

6. PLOS authors have the option to publish the peer review history of their article (what does this mean?). If published, this will include your full peer review and any attached files.

Reviewer #1: **Yes: **Ana Doblas

Reviewer #2: No

---

## [Author Response · Author response to Decision Letter 0]

29 Jun 2023

We are grateful to the reviewers for their insightful comments. We have adjusted and improved the manuscript in response to these.

We have addressed each of the comments from the reviewers separately within an attached document ('Response_to_reviewers.docx'), this can be found at the bottom of the submission document. We hope that we have addressed the reviewers concerns sufficiently and would like to thank them for taking the time to improve our manuscript.

---

## [Decision Letter · Decision Letter 1]

10 Aug 2023

PONE-D-23-11327R1SimpliPyTEM: An open-source Python library and app to simplify transmission electron microscopy and in situ-TEM image analysis.PLOS ONE

Dear Dr. Ing,

Thank you for submitting your manuscript to PLOS ONE. After careful consideration, we feel that it has merit but does not fully meet PLOS ONE’s publication criteria as it currently stands. Therefore, we invite you to submit a revised version of the manuscript that addresses the points raised during the review process. Please, consider and answer all the reviewers concerns and requests in order to improve the quality of your submission.

We look forward to receiving your revised manuscript.

Kind regards,

Carlos Fernandez-Lozano, Ph.D

Academic Editor

PLOS ONE

Additional Editor Comments:

Please consider the reviewer comments in order to improve your paper.

Reviewers' comments:

Reviewer's Responses to Questions

**Comments to the Author**

1. If the authors have adequately addressed your comments raised in a previous round of review and you feel that this manuscript is now acceptable for publication, you may indicate that here to bypass the “Comments to the Author” section, enter your conflict of interest statement in the “Confidential to Editor” section, and submit your "Accept" recommendation.

Reviewer #2: (No Response)

2. Is the manuscript technically sound, and do the data support the conclusions?

Reviewer #2: Partly

3. Has the statistical analysis been performed appropriately and rigorously? 

Reviewer #2: N/A

4. Have the authors made all data underlying the findings in their manuscript fully available?

Reviewer #2: Yes

5. Is the manuscript presented in an intelligible fashion and written in standard English?

Reviewer #2: Yes

6. Review Comments to the Author

Reviewer #2: The authors had done a reasonable job of revising this manuscript based on the reviews. However, there remain some significant issues. Before I can evaluate the overall ms however, the program needs to be able to run without being edited.

I remain unable to get the GUI program (as posted on GitHub) to run, which means that there is something about the structure or instructions that is not correct. Following the instructions on readthedocs, under "SimpliPyTEM-GUI", an attempt to run the program throws the error "ModuleNotFoundError: No module named 'SimpliPyTEM'". I suggest the authors try an install on a clean machine (no pre-existing conda library or pip installs), and be sure that the instructions result in a runable GUI. I also suggest including the install instructions clearly in the README.

There are 2 ways in which users will obtain this program: either directly from the github repo, or from pypi. The instructions should be clearly separated to be able to make working environments. I note that this applied to using the GUI; I suspect that using the library programically is fine however.

I finally did get the program to come up by removing the "SimpliPyTEM." from all the imports in all of the source files when running from the terminal, and changing the code a little.

After that, I attempted to load a simple tiff file with the GUI. The call to "process_file" complained that there were 10 position arguments, but 12 were given in the call (this is all within SimpliPyTEM_GUI.py). The call specifies self.topaz_on and self.cuda_on, but these are not in the argument list for the function itself.

I suggest several things:

1. Use mypy or another linter to seriously lint the program. The GUI part at least is not ready for release.

2. Use typed arguments (hints). You may have to move up from python3.8 to 3.10 or 3.11 to fully take advantage of this. However, it would avoid the error above where the function call and the arguments did not agree. It also will be sure that the right argument is passed to the right variable in the function call, and can define values for variables that are not passed (solving the problem above).

3. The color scheme did not work well on my computer. The text in the GUI was white, the background was light cyanish, and the text in the text boxes was also white and unreadable. My computer runs in "dark mode" so text is typically white or light, but attempts to change the style within the GUI (e.g., settting to "fusion", which works with other code) did not work to make the text stand out. I am not sure the add_styles function was accomplishing its goals either.

Some specific minor comments:

Line 86: , -> .

line 86: Python is not always faster than ImageJ. ImageJ can be scripted (even with Python) to automate tasks - not easy, but not hard either. I would modify this statement.

line 93: "app" is jargon. Use either "application" or "program" to refer to the GUI, or "library" for the library functions.

lien 294: "PyFPDF" is no longer maintained according to their webpage. Perhaps use the replacement?

7. PLOS authors have the option to publish the peer review history of their article (what does this mean?). If published, this will include your full peer review and any attached files.

Reviewer #2: No

---

## [Author Response · Author response to Decision Letter 1]

30 Aug 2023

We would like to thank the reviewers for taking the time to give feedback on our manuscript. We have provided a response to each of the reviewer comments within an attached document which can be found at the end of this submission document. We hope that this document has adequately addressed the reviewers' concerns.

---

## [Decision Letter · Decision Letter 2]

19 Sep 2023

SimpliPyTEM: An open-source Python library and app to simplify transmission electron microscopy and in situ-TEM image analysis.

PONE-D-23-11327R2

Dear Dr. Ing,

We’re pleased to inform you that your manuscript has been judged scientifically suitable for publication and will be formally accepted for publication once it meets all outstanding technical requirements.

Kind regards,

Carlos Fernandez-Lozano, Ph.D

Academic Editor

PLOS ONE

Additional Editor Comments (optional):

I wanted to extend my sincere appreciation for the diligent efforts you put into addressing the comments and suggestions provided by the reviewers for your article titled. Your attentiveness to their feedback and your subsequent revisions have significantly enhanced the quality and scholarly merit of the manuscript.

Reviewers' comments:

Reviewer's Responses to Questions

**Comments to the Author**

1. If the authors have adequately addressed your comments raised in a previous round of review and you feel that this manuscript is now acceptable for publication, you may indicate that here to bypass the “Comments to the Author” section, enter your conflict of interest statement in the “Confidential to Editor” section, and submit your "Accept" recommendation.

Reviewer #2: All comments have been addressed

2. Is the manuscript technically sound, and do the data support the conclusions?

Reviewer #2: Yes

3. Has the statistical analysis been performed appropriately and rigorously? 

Reviewer #2: N/A

4. Have the authors made all data underlying the findings in their manuscript fully available?

Reviewer #2: Yes

5. Is the manuscript presented in an intelligible fashion and written in standard English?

Reviewer #2: Yes

6. Review Comments to the Author

Reviewer #2: With the new instructions, I was able to get the program running and do some simple operations with it. My previous concerns have all been adequately addressed.

There was only one minor issue, which is that imagecodecs (Gohlke's library) is no longer compatible with Python 3.8, so I had to bump the version to 3.9 to get the program to install, but once that was done the installation went smoothly. This is with the current versions and a Conda-built environment. The instructions might need to be updated to reflect this.

7. PLOS authors have the option to publish the peer review history of their article (what does this mean?). If published, this will include your full peer review and any attached files.

Reviewer #2: No

---

## [Editor Report · Acceptance letter]

26 Sep 2023

PONE-D-23-11327R2 

SimpliPyTEM: An open-source Python library and app to simplify transmission electron microscopy and *in situ*-TEM image analysis. 

Dear Dr. Ing:

I'm pleased to inform you that your manuscript has been deemed suitable for publication in PLOS ONE. Congratulations! Your manuscript is now with our production department. 

Kind regards, 

on behalf of

Dr. Carlos Fernandez-Lozano 

Academic Editor

PLOS ONE